# Comprehensive Development of a Cellulose Acetate and Soy Protein-Based Scaffold for Nerve Regeneration

**DOI:** 10.3390/polym16020216

**Published:** 2024-01-12

**Authors:** Brandon Gutiérrez, María Eugenia González-Quijón, Paulina Martínez-Rodríguez, Josefa Alarcón-Apablaza, Karina Godoy, Diego Pulzatto Cury, María Florencia Lezcano, Daniel Vargas-Chávez, Fernando José Dias

**Affiliations:** 1Master Program in Dental Sciences, Dental School, Universidad de La Frontera, Temuco 4780000, Chile; brandon.gutierrez@ufrontera.cl; 2Department of Chemical Engineering, Universidad de La Frontera, Temuco 4780000, Chile; mariaeugenia.gonzalez@ufrontera.cl; 3Scientific and Technological Bioresource Nucleus (BIOREN), Universidad de La Frontera, Temuco 4780000, Chile; karina.godoy@ufrontera.cl; 4Oral Biology Research Centre (CIBO-UFRO), Dental School, Universidad de La Frontera, Temuco 4780000, Chile; paulinaconstanza.martinez@ufrontera.cl; 5Research Centre in Dental Sciences (CICO-UFRO), Dental School, Universidad de La Frontera, Temuco 4780000, Chile; josefa.alarcon@ufrontera.cl; 6Doctoral Program in Morphological Sciences, Faculty of Medicine, Universidad de La Frontera, Temuco 4780000, Chile; daniel.vargas@umayor.cl; 7Department of Anatomy, Institute of Biomedical Sciences, Universidade de São Paulo (ICB-USP), São Paulo 05508-000, Brazil; diegocury@usp.br; 8Department of Cellular Biology and the Development, Institute of Biomedical Sciences, Universidade de São Paulo (ICB-USP), São Paulo 05508-000, Brazil; 9Departamento de Bioingeniería, Facultad de Ingeniería, Universidad Nacional de Entre Ríos, Oro Verde 3100, Argentina; lezcano.f@gmail.com; 10Facultad de Medicina y Ciencias de la Salud, Universidad Mayor, Escuela Medicina Veterinaria, Temuco 4780000, Chile; 11Department of Integral Adults Dentistry, Dental School, Universidad de La Frontera, Temuco 4780000, Chile

**Keywords:** nerve injury, biomaterial, regenerative biology, nerve guide conductors, biocompatibility

## Abstract

Background: The elaboration of biocompatible nerve guide conduits (NGCs) has been studied in recent years as a treatment for total nerve rupture lesions (axonotmesis). Different natural polymers have been used in these studies, including cellulose associated with soy protein. The purpose of this report was to describe manufacturing NGCs suitable for nerve regeneration using the method of dip coating and evaporation of solvent with cellulose acetate (CA) functionalized with soy protein acid hydrolysate (SPAH). Methods: The manufacturing method and bacterial control precautions for the CA/SPAH NGCs were described. The structure of the NGCs was analyzed under a scanning electron microscope (SEM); porosity was analyzed with a degassing method using a porosimeter. Schwann cell (SCL 4.1/F7) biocompatibility of cell-seeded nerve guide conduits was evaluated with the MTT assay. Results: The method employed allowed an easy elaboration and customization of NGCs, free of bacteria, with pores in the internal surface, and the uniform wall thickness allowed manipulation, which showed flexibility; additionally, the sample was suturable. The NGCs showed initial biocompatibility with Schwann cells, revealing cells adhered to the NGC structure after 5 days. Conclusions: The fabricated CA/SPAH NGCs showed adequate features to be used for peripheral nerve regeneration studies. Future reports are necessary to discuss the ideal concentration of CA and SPAH and the mechanical and physicochemical properties of this biomaterial.

## 1. Introduction

Injuries of the peripheral nerves can occur due to ischemic, chemical, mechanical, and thermal factors or during surgery [1], which leads to the partial or complete loss of sensory, motor, and autonomous functions and the generation of neuropathic pain, which can negatively affect quality of life [1,2]. Many people are affected by peripheral nerve injuries; according to recent studies, the mean incidence of this type of injury in England is 11.2 events per 100,000 persons per year [1] and 7.8 events per 100,000 persons per year in South Korea [2]. In developed countries, 13 to 23 of 100,000 persons/year are affected by nerve injuries [3] and more than 100,000 patients undergo nerve repair surgeries in the United States and Europe every year [4].

Nerve injuries are divided into neuropraxia, axonotmesis, and neurotmesis [1,5,6]. There is no loss of continuity in the nerve trunk in neuropraxia and axonotmesis, so it can regenerate spontaneously [7,8]; in the presence of neurotmesis, the loss of continuity of the nerve trunk occurs [1,5,6], and the length of the nerve gap or the distance between the proximal and distal ends affects the capacity for functional recovery [5,9,10]; in these cases, functional recovery is rarely restored [7,8], therefore patients do not recover normal motor control and fine sensitivity [11].

Treatment methods for nerve injuries can be categorized into surgical and nonsurgical treatments, depending on the type of nerve injury [5]. Surgical techniques are most frequently used in the treatment of neurotmesis [1], and the “gold standard” in the treatment of neurotmesis is autologous grafts [1,2,3,8,12,13,14,15]; however, the success rate of this therapy does not always result in correct nerve regeneration [14,16] and is associated with loss of function [1,3,10,14] and potential neuroma formation at the donor site [14,15]. To overcome these disadvantages, the development of NGCs to unite defects in peripheral nerves has been an important field of research in recent years [2,3,14,15,17].

NGCs are tubular scaffolds through which the regeneration of the lost nervous tissue is stimulated and directed [9]. The requirements of an ideal NGC include an architecture of mechanical properties that provide structural support, permeability, conductivity, flexibility, and biodegradability [1,3,13]. The design of NGCs is varied, and the configuration of the fibers, presence of pores and grooves, and thickness of the walls are fundamental to achieve successful nerve regeneration [1,13,18].

Several manufacturing methods have been described for the elaboration of NGCs [1,19], namely, drying in the cold (freeze-drying), micropatterning, electrospinning, additive manufacturing, gas foaming, phase separation, and finally evaporation, for which there are three different methods, namely, the dip–coating method, molding method (solvent casting), and rolling method; the latter can be used both for the solvent evaporation method and micropatterning and electrospinning. Each one has advantages and disadvantages that need to be investigated [1]. The solvent evaporation technique is described as a simple, reproducible, and low-cost procedure, but its main drawbacks are the use of toxic chemical solvents and irregular porosity [1].

There is a large list of biomaterials used in the creation of an NGC applied to nerve regeneration [1,3,8,12,13,14]. Nondegradable synthetic materials, such as silicone, were the first materials to be used [1,14] but have been abandoned, as they can cause a chronic reaction to a foreign body and mechanical grip, which make functional recovery difficult and a second intervention for the removal of these materials necessary [20,21]. This is the reason why biodegradable polymeric materials have been used to manufacture NGCs [1,9,13,17,21].

Natural polymer materials such as collagen [1,15,22], chitosan [15,23], silk [15,22,24], cellulose, and soy protein [1,2,21,25] have demonstrated great efficiency in nerve regeneration comparable to autologous nerve grafts [21]. Among the natural polymers, cellulose is the most abundant [21,26], and its biocompatibility, biodegradability, and mechanical properties have already been studied and adapted to different fields of tissue engineering [2,12,21,25]. In recent years, the association of cellulose/isolated soy protein (ISP) has been shown to be effective in the creation of NGCs capable of promoting nerve regeneration in sciatic nerve defects measuring 10 mm in width in rats [21], and ISP scaffolds have revealed biodegradable and biocompatible properties [2,12,21,25,26] and cytocompatibility with Schwann cells [21,27]; in addition, soy protein provides a morphological functionalization of porosity to the scaffold [28], which provides permeability allowing the supply of nutrients and exchange of gases and could prevent scar tissue infiltration and the diffusion of growth factors outside the lumen [13,29].

The purpose of this report is to describe the details and precautions for manufacturing NGCs with morphological and structural characteristics suitable for nerve regeneration using the method of dip coating and evaporation of solvent with cellulose functionalized with soy protein. The NGC structure is analyzed via scanning electron microscopy; its porosity via porosimeter; and its biocompatibility with Schwann cells via MTT assay and scanning electron microscopy.

## 2. Materials and Methods

Cellulose acetate (CA) (Product Code: 419028, ~50 g/mol, Sigma-Aldrich, Saint Louis, MO, USA) and soy protein acid hydrolysate (SPAH) (Product Code: S1674, Sigma-Aldrich, Saint Louis, MO, USA) were used to manufacture the NGCs. Analytical grade acetone (Product Code: 270725 purity 99.9%, 58.08 g/mol, Sigma-Aldrich, Saint Louis, MO, USA) was used to dissolve the CA and SPAH.

Prior to use, the CA was sterilized in 0.5 g aliquots in 2 mL cryogenic tubes (Sorenson BioScience, Salt Lake City, UT, USA). Before use, the SPAH was subjected to ultraviolet radiation for 30 min inside the laminar flow cabinet of a vertical air blower (BBS-v800, Biobase, Tsinan, China).

The antisepsis of the work area was carried out with ethyl alcohol 70° (Pure Ethyl Alcohol, Sigma-Aldrich, Saint Louis, MO, USA). The necessary instruments were as follows: anatomical forceps, scalpel handle No. 3, disposable scalpel blade No. 15, 1.4 mm diameter galvanized wire, 6 cm straight scissors, disposable sterile field cloths (square area of 0.25 m^2^) and sterile gauze (square area of 0.0016 m^2^). The asepsis of the instruments and aliquots of CA was carried out in a 12-L semiautomatic autoclave (Dabi-Atlante, Ribeirão Preto, Brazil).

### 2.1. Protocol for the Elaboration of NGC Cellulose Acetate/Soy Protein Acid Hydrolysate

Prior to the elaboration of the NGC, the asepsis and antisepsis of the instruments, supplies, and work area were carried out; all metallic instruments, beakers, magnetic stirrers, and cellulose acetate were sterilized; the antisepsis of the work area was carried out with 70° alcohol, and it was subjected to UV radiation for 30 min in a vertical laminar flow chamber. The NGC manufacturing protocol is described below.

In a sterile beaker, CA was dissolved in acetone, at concentrations of 9.08, 11.3, and 14.4 wt%; the solution was mixed for 10 min with a sterile magnetic stirrer at room temperature until a clear and homogeneous solution was achieved.In parallel, in a sterile beaker, the SPAH was dissolved in acetone at a concentration of 25 and 30 wt%; the mixture was stirred for 30 min at room temperature until a homogeneous solution was achieved under constant UV light irradiation.The SPAH solution was slowly poured into the CA solution, stirring constantly for 15 min at room temperature under UV light irradiation until a homogeneous and bubble-free mixture was achieved. The final concentrations were as follows: S1, CA: 4.1 and SPAH: 16.3 wt%; S2, CA: 5.1 and SPAH: 13.6 wt%; and S3, CA: 6.5 and SPAH: 13.6 wt%.The solution was deposited in 15 mL conical-bottom tubes/graduated cylinder (Corning, Corning, NY, USA), and before it settled, a 1.4 mm diameter galvanized rod/wire was introduced for 5 s and slowly withdrawn while making rotational movements so the polymer excess dropped off. In this way, the tubular shape of the NGC was obtained (Figure 1A).The above procedure was repeated 9 times, always letting it drain between each immersion. The new immersions were carried out 60 s apart to allow the complete evaporation of the acetone and integration of the layers.Finally, the rod with the impregnated polymers was placed in ultrapure distilled water for 5 min.With the help of scissors, the ends were cut, and the NGC was removed by pushing and rotating at the same time (Figure 1B,C).Disinfection protocol: the NGCs were submerged in 70° alcohol in 2 cycles of 30 min each (Figure 1D).Subsequently, the NGCs were preserved in a 1% penicillin/streptomycin antibiotic solution.

### 2.2. NGC Microbiological Growth Tests

To determine the adequate elaboration protocol described, it was necessary to test different methodologies to ensure the sterility of the NGCs. The NGCs were manufactured following the same manufacturing steps considering the following variables: the method of asepsis/antisepsis of CA and PS polymers and the medium for maintenance or cultivation.

To determine the adequate sterility protocol for preparing NGCs, a total of 16 NGCs were manufactured, of which 8 were made with autoclaved CA and sterilized SPAH under UV radiation for 30 min, and 8 were prepared with CA and PS sterilized under UV radiation for 30 min. Half of the samples from each group were cultured for 7 days at room temperature; additionally, half of the samples (*n* = 4) were cultured in 10 mL ultrapure distilled water (Corning), and the other half (*n* = 4) were cultured in 1% penicillin/ streptomycin (Cytiva, Marlborough, MA, USA).

After 7 days, longitudinally cut 6 mm long NGC sections were cultured for 48 h in Ham’s F12 culture medium (Cytiva) supplemented with 10% fetal bovine serum (Cytiva, 37 °C, 5% CO_2_ in a culture chamber). Sample analysis was initially performed using an inverted microscope (EUROMEX), observing mainly the characteristics of the culture medium. After verifying the absence of supernatant material, the analysis of the NGC samples was completed using a scanning electron microscope (SEM-Hitachi SU3500, Hitachi, Tokyo, Japan).

### 2.3. NGC Structural Characterization—Macroscopic and Scanning Electron Microscopy Analyses

The morphological and morphometric analysis of the manufactured NGCs was performed with a scanning electron microscope (SEM-Hitachi SU3500, Hitachi, Tokyo, Japan) at a voltage of 10 kV. The samples were cut longitudinally and transversally. The samples were observed at different magnifications to analyze the external face, internal face, and cross-section. The morphometric structural characterization of the images was performed using the ImageJ v1.46r software (National Institutes of Health, Bethesda, MD, USA).

### 2.4. NGC Porosity and Surface Area—Nitrogen Adsorption and Degassing

The analysis of the surface area and pore size was performed using a porosimeter device (NOVA 1000e, Quantachrome instruments, Boynton Beach, FL, USA) according to a nitrogen adsorption and degassing method. Prior to analysis, the NGC samples were dried in a constant temperature incubator (Biobase, Tsinan, China) at 50° for 24 h. The results delivered were pore volume (g/mm^3^), surface area (mean unit), and pore diameter (measured unit). The results were analyzed with Software Design Expert 6.0.6 (Stat-ease, Inc., Minneapolis, MN, USA).

### 2.5. Schwann Cell Viability Direct Biocompatibility—MTT Assay and Scanning Electron Microscopy

In vitro biocompatibility tests were carried out with a colorimetric assay to evaluate cellular metabolic activity according to the MTT assay.

Schwann cells (SCL 4.1/F7, ECACC 93031204) were grown in a T-75 bottle (Corning, Corning, NY, USA) with Ham’s F12 culture medium (Cytiva) supplemented with 10% fetal calf serum (Cytiva) and 1% penicillin/streptomycin (Cytiva) at 37 °C, 5% CO_2_ in a humidified environment (culture chamber). Cells were released from the plate with 2 mL of 0.25% Trypsin-EDTA (Cytiva) at 37 °C, 5% CO_2_ in a humidified environment. After incubation, 4 mL of supplemented culture medium was added to inactivate the enzyme. Cells were collected in a 15 mL conical bottom tube (Corning) and centrifuged at 1000 rpm for 5 min. Cells were suspended in Ham’s F12 medium supplemented at a concentration of 100,000 cells/10 µL.

Longitudinally cut, 6 mm long NGCs were placed in a 96-well plate (Corning) to which 100,000 cells were seeded on the inside of the NGC; the plates were incubated for 2 h at 37 °C, 5% CO_2_ in a humidified environment. At the end of the incubation, 90 µL of supplemented culture medium was added to the wells until the next test.

The MTT test (Proliferation Kit I, Roche, Basel, Switzerland) was applied to assess each NGC at 24 and 48 h [30]. NGCs were transferred to a new 96-well plate supplemented with 100 µL of Ham’s F12, and 10 µL of “MTT labeling reagent, 1×” (final concentration 0.5 mg/mL) was added to each NGC-containing well. The plates were incubated for 4 h at 37 °C, 5% CO_2_. Once the incubation was finished, 100 µL of the “solubilization buffer, 1×” reagent was added, and the plates were incubated overnight at 37 °C, 5% CO_2_. The following day, the NGCs were removed, and the reading was performed using Infinite50 equipment (Tecan, Männedorf, Switzerland) at 570 nm. The data were collected and analyzed in Excel (Office 365 Education—Microsoft, Redmond, WA, USA).

A total of 100,000 Schwann cells were seeded on the inside of longitudinally cut, 6 mm long NGCs, which were incubated for 24 and 72 h at 37 °C, 5% CO_2_. As soon as the incubation time was over, the NGCs were washed twice with 1 mL of 1× PBS (Cytiva), and 2 mL of 2.5% glutaraldehyde in 0.1 M Sorensen’s phosphate buffer (Electron Microscopy Sciences—EMS, Hatfield, PA, USA) was added. The samples were incubated for 30 min at room temperature. The fixed NGCs were washed 3 times with ultrapure distilled water (Corning) and analyzed under a scanning electron microscope (SEM-Hitachi SU3500, Hitachi, Tokyo, Japan) at a voltage of 10 kV.

### 2.6. Statistical Analysis

Statistical analysis of data was performed using SigmaPlot 12.0 software (Systat Software Inc., Palo Alto, CA, USA). Quantitative data are presented as the mean ± standard deviation. Statistical analysis was performed with a one-way analysis of variance (ANOVA), followed by a Bonferroni post-test. Values of *p* < 0.05 were considered statistically significant.

## 3. Results

### 3.1. NGC Microbiological Growth Tests

Observation of the Ham’s F12 culture media under the inverted microscope after the 48 h NGC incubation indicated that bacteria were only found in the samples prepared with CA and PS irradiated with UV light and submerged in ultrapure distilled water for the previous 7 days. The other three preparation protocols did not reveal characteristics that would suggest contamination and bacterial proliferation in the prepared samples. Thus, the samples were analyzed using a scanning electron microscope to confirm the absence of bacteria (Figure 2A).

The analysis of NGCs under SEM showed that bacteria adhered to the NGC structure of the samples prepared with CA and PS irradiated with UV light regardless of if they were immersed in ultrapure distilled water or 1% penicillin/streptomycin for 7 days; the strains found corresponded to diplococci and bacilli (Figure 2B). However, no bacteria were adhered to the NGC structures fabricated with autoclaved/sterile CA and PS irradiated with UV light, regardless of the medium in which they were immersed (Figure 2C).

### 3.2. NGC Structural Characterization

#### 3.2.1. NGC Porosity and Surface Area—Liquid Nitrogen Adsorption and Degassing

The analysis of the surface area and porosity of NGC samples prepared with different CA and SPAH concentrations using the porosimeter with the liquid nitrogen adsorption and degassing method revealed the following data presented in Table 1.

Considering the porosity assessment, it was determined that all following analyses were carried out on the NGC samples prepared with a concentration of CA5.1% and SPAH 13.6% wt, as they presented the largest surface area and pore volume.

#### 3.2.2. Macroscopic Structural Characterization

Using the proposed manufacturing protocol, it was possible to manufacture NGCs up to 50 mm in length, which allowed cutting to different lengths according to different needs. These NGCs revealed consistency and flexibility similar to cardboard in samples that were kept moist (Figure 3A). The internal diameter remained constant at 1.4 mm, which corresponded to the diameter of the steel rod used as the mold (Figure 3B). This size was defined as ideal for implantation in the sciatic nerve of rats, but the size can also be customized depending on the nerve in which it is intended to be implanted.

The NGC structure obtained was suturable (Figure 3C), granting resistance to tearing using silk sutures and polyglactin 910. In addition, the tubular structure offered resistance to compression and some flexibility when pressed with the fingers and taken with anatomical forceps. This would allow its manipulation and implantation in future animal studies.

#### 3.2.3. Microscopic Structural Characterization—Scanning Electron Microscopy

Analysis of CA and SPAH NGCs subjected to different sterilization and storage protocols revealed different morphological characteristics observed under SEM (Figure 4). It was noted that compared to the NGCs that were manufactured from cellulose subjected to the UV irradiation sterilization process, those manufactured from cellulose subjected to the autoclave sterilization process presented an internal surface with a rougher structure (Figure 5F,H) and better distribution of an apparently greater number of pores of different sizes (Figure 5B,D). The medium in which the samples were stored, regardless of the sterilization method, did not seem to alter the structure of the manufactured tubular scaffolds. In the four variations, the wall of the manufactured NGCs did not appear as a compact structure, revealing spaces interspersed with a trabecular structure.

#### 3.2.4. Microscopic NGC Morphometrics—Scanning Electron Microscopy

The morphometric characterization of the structure of the NGCs manufactured with the four sterilization and storage protocols was carried out using the SEM images obtained for the samples. The thickness of the walls and the diameter of the pores observed on the inner surface were analyzed and compared. In order to improve the analysis, the evaluated pores were divided into two types: major pores (>25 μm) and minor pores (<25 μm).

Table 2, Table 3 and Table 4 reveal the data for the NGC wall thickness and the major/minor internal pores, respectively.

Morphometric analysis of the NGCs manufactured with the four different sterilization and storage protocols revealed no significant differences (*p* > 0.05) in relation to the thickness of the walls of the tubular structure. The NGC wall thicknesses ranged between 237.5 and 275.8 µm.

Analysis of observable pores revealed significant differences in major pores. NGCs prepared with CA and SPAH irradiated with UV and stored in ultrapure water revealed the largest major pores (61.1 ± 5.4 µm, *p* < 0.05) compared to all other sterilization and storage protocols. Furthermore, samples of NGCs prepared with CA and SPAH irradiated with UV and stored in 1% penicillin/streptomycin revealed significantly smaller major pores (32.8 ± 3.7 µm, *p* < 0.05) than NGCs prepared with autoclaved CA and UV-irradiated SPAH stored in ultrapure water.

The minor pores also did not reveal significant differences (*p* > 0.05) among the four sterilization and storage protocols. The minor pore values for the NGCs ranged between 2 and 23.5 µm.

### 3.3. MTT Assay—Schwann Cell Viability/Biocompatibility

Table 5 and Table 6 show the results of the first and second replicates of the cell viability assay, respectively.

Biocompatibility analysis using the MTT assay revealed a similar Schwann cell viability after 48 h in the two replicates performed on NGCs prepared with cellulose sterilized by autoclaving and the soy protein/cellulose mixture treated with UV light and stored in medium with 1% penicillin/streptomycin. Cell viability values ranged from 88.7% to 60.96%, with the average of the three samples from the first replication being 68.75% and that for the second replication being 69.52%.

### 3.4. NGC Direct Cytocompatibility of Schwann Cells

Samples of cellulose NGC functionalized with soy protein, analyzed using a scanning electron microscope without the seeded Schwann cells, revealed a porous structure without the presence of bacteria, as previously described.

The direct cytocompatibility analysis of Schwann cells was performed for three different periods, namely, 1, 3, and 5 days after cell seeding on the prepared NGCs (Figure 5). The different time points revealed different characteristics.

After 1 day of Schwann cell seeding, a monolayer of Schwann cells was observed that proliferated and adhered to the internal face of the tubular scaffold. These Schwann cells revealed cytoplasm with multiple projections with starry morphology occupying most of the surface, except for areas with pores larger than 60 µm; in these spots, the cells were not capable of forming a uniform layer.

In this period of analysis, some spherical structures were also observed, which suggested the presence of cells not adhered to the structure of the NGC because they were on top of other adhered cells, a fact that reinforced the formation of a monolayer of cells adhered to the internal surface of the manufactured cellulose NGCs.

After 3 and 5 days of cell seeding, the presence of Schwann cells adhered to the internal surface of the NGCs was still observed. Nevertheless, a reduction in the number of cells and the area occupied by the monolayer on the inner surface of the tubular scaffold was evident. The Schwann cells observed in these periods were adhered to the scaffolds, showing long cytoplasmic extensions and, consequently, maintaining the star-like shape.

## 4. Discussion

There are currently several methods for preparing NGCs [1,19]; however, the detailed manufacturing protocols, which would allow the exact reproducibility of these scaffolds, are normally not published to safeguard the copyright information.

Nevertheless, this report aimed to explain the entire manufacturing process of an NGC in a simplified and understandable manner, favoring its reproducibility and exposing some problems and solutions found during its development, such as the tube molding process and bacterial contamination.

The NGC tube molding was achieved after trying three variations of the solvent evaporation technique: 1. the solvent casting method, 2. the rolling layer method, and 3. the dip-coating method [1,21].

I.Solvent casting (molding method) used cellulose/soy protein dissolved in urea/NaOH solution and poured into a polyurethane mold with a central glass rod; the coagulation was carried out with the application of acetic acid [23,31]. Another study used hydroxyethyl cellulose and isolated soy protein dissolved in NaOH solution, and the solution was poured into molds with 1.3 or 1.7 metal rods; through a freeze-drying process, the final tubular shape was achieved [28].

In our methodological proposal, it was not possible to achieve an adequate resistant tubular structure; coagulation at room temperature results in a gel form that is complex to demold a tubular structure. The cellulose acetate was insoluble in a solution of urea/NaOH; thus, an attempt was made to coagulate the polymer solution with 5% acetic acid without obtaining positive results. In addition, the tubular structure adhered to the mold, which prevented its demolding. These inconveniences led us to discard this evaporation method.

An attempt was made to use petroleum jelly on the surfaces of the mold and central rod as well as to accelerate the acetone evaporation process in a constant temperature incubator (Biobase, Tsinan, China) at 25 °C for 5–10 and 15 min. The result was an amorphous and fragile NGC due to syneresis by polymer baking. In addition, it was suggested that the petroleum jelly adhered to the NGC could interfere with the cell adhesion results, and no literature was found using petroleum jelly or methods for its removal from the tubular structure (Figure 6A). These inconveniences led us to discard this evaporation method.

II.Rolling layer method (coiled membrane, Figure 6B–D): This method was used before to manufacture cellulose/soy protein membranes [31]. To give it a tubular shape, the floating membrane in distilled water was taken with tweezers, and with the help of a metallic rod with a 1.4 mm diameter, it was rolled by pressing on a sterile gauze or gauze pad; the rolling method is complex, and three disadvantages were detected:
1.The membrane, still wet, is fragile, and therefore cracks and ruptures of the membrane occur, which are difficult to prevent during handling.2.The thickness of the floating membrane cannot be estimated, so it is difficult to predict the thickness of the resulting NGC walls; in addition, water and acetone are trapped between each membrane twist.3.As the soy protein is hydrophilic and disintegrates into islets it negatively affects the final morphology and porosity of the NGCs.

A second way to use this method of membrane elaboration is to let the polymers rest with the solvent in a Petri dish (Sorenson BioScience, Salt Lake City, UT, USA) for 24 h at room temperature [26] achieving the evaporation of the acetone without adding water. The results are not positive, possibly due to the different molecular weights of the polymers, which generate the decantation of the soy protein, obtaining a cellulose acetate membrane with a plastic aspect that is not flexible, is brittle, and is impossible to roll up. The disadvantages raised led us to discard this method.

III.Dip-coating method: After the failure of the aforementioned methods, the idea of granting the tubular shape by introducing a rod into the polymer solution [21] arose. When it was withdrawn intermittently from the solution, the evaporation of the solvent occurred, leaving the mixed polymers adhered to the rod; successively, in this way, the thickness of the NGC can be increased in a controlled manner. The process is practical and simple, allowing us to develop an NGC with a determined thickness and internal diameter; in addition, the resulting NGC presents relative flexibility, an ideal morphological characteristic that an NGC should possess [15]. Currently, the dip-coating manufacturing method has already been used in the manufacture of NGCs [21]; however, this process has not been reported in the literature when used with cellulose acetate functionalized with soy protein. A systematic review [21] reported that a disadvantage of the dip-coating method for the manufacture of NGCs is the impossibility of creating pores for the transport of nutrients. Our study, however, showed that the manufactured NGCs had pores on the inner face, in addition to a porous structure on the wall of the tubular structure.

The solvent evaporation method eliminated the disadvantages of the two aforementioned methods. When the rod was introduced into the polymers and slowly removed, a thin layer of cellulose/soybean protein was adhered, which allowed controlling the final thickness of the NGC; no oven was required to accelerate the evaporation of the solvent; the tubular structure formed instantaneously without requiring rolling; by not letting the polymers rest, there is no decantation and/or separation due to their different molecular weights; and the tubular structure was adhered only by the walls of the lumen facilitating the removal of the metal rod.

### 4.1. Concentrations of Cellulose Acetate and Soy Protein for NGC Manufacturing

The literature reveals that the concentrations of cellulose acetate and soy protein suitable for the manufacture of NGCs have not yet been defined since studies have reported a great variety of concentrations. Luo et al. [31] and Gan et al. [23] used a concentration of 3.5% soy protein and 10% cellulose; Kim et al. [32] used between 5 and 20 wt% of soy protein associated with poly(d,l-lactide-co-glycolide) to prepare microcapsules. Wang et al. [26] in chitosan/SPI membranes used a concentration of 3 wt% chitosan and varied between 0 and 30 wt% soy protein.

It has been determined that the SPAH concentration is the variable that determines the integrity and porosity of NGCs [26,31].

In our research, by mixing the independent solutions of each polymer, three different solutions, namely, CA 4.1 wt% and SPAH 16.3 wt%, CA 5.1 wt% and SPAH 13.6 wt%, and CA 6.5 wt% and SPAH 13.6 wt%, were obtained. It was noted that the different concentrations of CA and SPAH generated different porosities, in which both the increase in the CA and SPAH values and the reduction in the CA concentration revealed a reduction in the porosity of the samples obtained. For this reason, the sample with intermediate values of CA (5.1 wt%) and SPAH (13.6 wt%) was chosen, which revealed better porosity to carry out structural and biocompatibility analyses with Schwann cells. Furthermore, mixtures with CA concentrations above 6.45 wt% and SPAH above 17 wt% did not enable the manufacture of NGCs.

### 4.2. Bacterial Control of NGCs

NGC sterility is essential for biocompatibility with Schwann cells and subsequent implantation in living beings. After observing the first manufactured NGC samples under SEM, we found the presence of bacteria adhered to the structure.

Sarhane et al. [33] performed sterilization using ethanol solution baths (70° and 95° for 30 min) in scaffolds prepared by electrospinning poly-є-caprolactone. In our case, despite carrying out this process, it was not possible to eliminate the bacteria that adhered to the NGCs. Romo-Velera et al. [34] used UV light for sterilization of scaffolds made of collagen/soy protein, irradiating the membrane for 20 min. In the same way, Oh et al. [35] sterilized the NGCs of PLGA with ultraviolet irradiation for 30 min, and Varshney et al. [36] sterilized silk/soy protein membranes with UV light for 30 min. The drawback of adopting this sterilization medium is that the tubular structure of the NGC makes it difficult for UV radiation to penetrate the lumen.

Therefore, the sterilization of NGCs with steam in an autoclave was proposed by Luo et al. [31] for cellulose membranes and soy protein and by Kundu et al. [27] for silk fiber NGCs. Since our tubular structure of the CA/SPAH suffers syneresis, modifying its physical properties, this method of sterilization was also discarded.

Considering the previous background, it was decided to carry out the autoclave sterilization methods of the CA and PS polymers dispensed in 1 g aliquots, which were used in the manufacture of the NGCs.

In our experiments, it was possible to sterilize the CA using moist heat in the autoclave cycle, without syneresis or imbibition of the material. On the other hand, SPAH underwent water imbibition in the autoclave sterilization process, which transformed the material into a compact mass that was not possible to subsequently dissolve in acetone to generate the final mixture with CA.

Thus, it was decided to sterilize the SPAH with UV radiation for 45 min during its mixing phase with acetone. In addition, a storage medium with ultrapure distilled water with 1% penicillin/streptomycin was used, and an aqueous medium was used in cell culture [34,37]. With these precautions, the sterility of the developed CA/SPAH NGCs was achieved.

### 4.3. Structural Characteristics of NGCs

The elaboration method proposed in the present study generated a structure with the presence of visible pores on the inner face and in the cross-section of the NGC. The internal diameter obtained was variable according to the diameter of the metal rod used.

Our NGC wall widths ranged from 237.5 to 278.1 µm, values considered adequate [15] that would allow correct handling and relative flexibility. In addition, our samples proved to be suturable with polyglactin 910.

The internal face porosity analyzed under SEM of the NGC samples was different from NGC images obtained by Gan et al. [23], in which it was not possible to observe internal pores. The addition of SPI caused microprotrusions on the surface that increased the structural surface area, as in our study, the release of SPAH from the polymeric matrix of cellulose acetate caused porosity to the structure and increased its surface area [38]. An immersion in a soluble liquid for soy protein for at least 24 h was necessary to form a porous tubular structure, similar to that reported by Luo et al. [29].

The concentration of 13.6 wt% of SPI used generates pores between 2 and 66 μm in diameter, with the average of “smaller pores” being 10 μm in diameter, similar to that reported by Gan et al. [23]. With the dip-coating method, it was possible to manufacture NGCs up to 6 cm in length, obtaining a uniform wall thickness; this is essential for repairing peripheral nerve injuries of greater extension.

Zhao et al.’s [28] scaffold elaborated with hydroxyethyl-cellulose/SPI reportedly had porosity with diameters from 20 to 200 µm in diameter, pores much larger than those reported in our study. A smaller pore range, as observed in our samples, could ensure the correct diffusion of nutrients [2] reducing the risk of infiltrating scar tissue [15].

### 4.4. Biocompatibility Evaluation of the Tubular Scaffold

Finally, the biocompatibility of Schwann cells was analyzed on the developed CA and SPAH NGCs.

Schwann cells were chosen for this analysis because they are recognized as fundamental glial cells in peripheral nerve regeneration. These cells contribute to neuronal survival and were important in the discovery of the axonal pathway, and they orchestrate the structure of the nerve with blood vessels and connective layers, endo, peri, and epineurium, during development. SCs exhibit remarkable plasticity in the context of nerve injury, wherein they differentiate into reparative cells and command a regenerative response that promotes nerve repair [39].

In our study, we have demonstrated a cellular viability of approximately 70% of SCs seeded on NGCs after 48 h. The viability values we obtained were similar to those reported by other authors using other cell types: 72% viability of human corneal epithelial cells and fibroblasts on ISP/glycerol membranes at 48 h [34] and 82.1% viability of fibroblasts on silk/ISP membranes over 4 days of culture [38]. Zhao et al. [38] evaluated the viability of fibroblasts seeded on hydroxyethylcellulose/SPI membranes around 80% at 72 h, concluding that the polymers pose low toxicity and high viability. Finally, Luo et al. [31] evaluated the cytocompatibility of SCs on cellulose/SPI membranes, reporting values of 120% viability at 72 h, as it seems that the high concentration of SPI at 30 wt% could be the factor that benefited this result.

Cellular cytocompatibility analysis under SEM was carried out on polymer membranes associated with soy protein. Wang et al. [40] in chitosan/ISP membranes concluded that ISP hydrolysis was beneficial for the proliferation, growth, and adherence of L929 cells. Zhao et al. [38] evaluated the cell arrangement of fibroblasts on membranes of hydroxyethyl-cellulose/ISP, reporting the presence of uniformly distributed cells. These results were similar to those observed in our investigation, namely, that the Schwann cells adhered and proliferated in the structure in a single layer. Specifically, regarding SCs, Luo et al. [31] analyzed their direct compatibility on cellulose/SPI membranes, showing good adhesion and proliferation 72 h after seeding. Our results were similar, revealing the presence of adhered cells up to 5 days after seeding on the developed AC/SPAH NGCs.

The NGCs manufactured with the proposed method were a first step towards the generation of a biomaterial that will allow the optimization and future complementation of this material with other nerve regeneration strategies. Our initial results suggest that this biomaterial could be enriched with cells and therapeutic agents as well as the combination of other biocompatible materials and neurotrophic growth factors that could generate an effective microenvironment that will favor cell growth and the regeneration of nervous tissue, including peripheral nerve tissue.

Among the study limitations and opportunities for improvement, the authors mention the possibility of refining the manufacture of NGCs with the evaporation of solvent/dip-coating method, mainly regarding the concentrations of CA and SPAH, which could improve the physicochemical properties of NGCs. These improvements could allow future implantation in different animal models for the study of functional recovery associated with nerve regeneration. Furthermore, there is still a lack of biomechanical analyses of this biomaterial and other more detailed cell proliferation and imaging analyses, such as confocal microscopy, which will help to understand the 3D interaction of cells with this biomaterial. Our research group will continue working to better understand and evaluate the material obtained, in addition to seeking to implement improvements in these NGCs.

## 5. Conclusions

The solvent evaporation/dip-coating method allowed an easy elaboration and customization of NGCs of cellulose acetate supplemented with soy protein. It was demonstrated in this report that a tubular structure could be fabricated, with different dimensions in length and internal diameter as well as presenting a uniform wall thickness that allowed its manipulation, which showed flexibility and was suturable.

The presence of pores on the surface and structural porosity was essential to allow the diffusion of nutrients. Initial biocompatibility analyses of NGCs with Schwann cells were promising. However, more cell proliferation analyses will be necessary to prove this biocompatibility.

In subsequent studies, it will be essential to find the ideal concentration of polymers to allow a higher concentration of SPAH since this is the polymer that provides porosity and increases biocompatibility. This study demonstrates that the NGC produced is safe and free of bacteria and that in the future it can be used for peripheral nerve regeneration studies as well as analyses of the mechanical and physicochemical properties of this biomaterial.

## Figures and Tables

**Figure 1 polymers-16-00216-f001:**
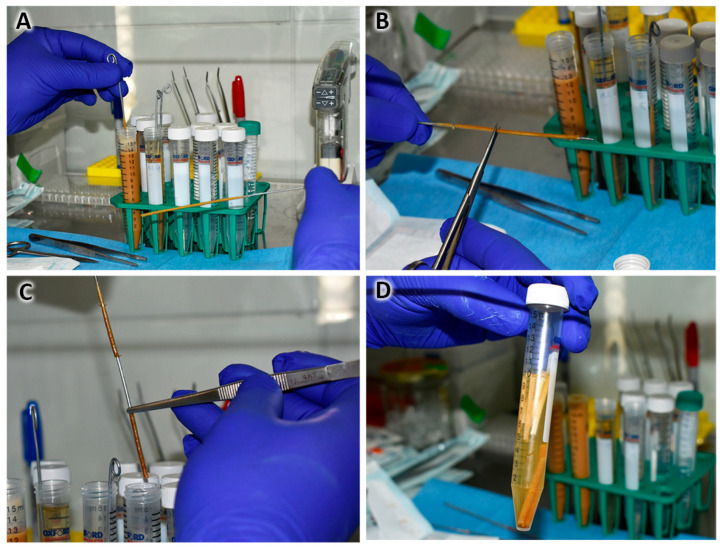
Steps in the method of preparing NGCs from cellulose and soy protein in a sterile environment (laminar flow chamber). (**A**) Immersions in 5.1% cellulose acetate solution and 13.6% soy protein acid hydrolysate. (**B**) NGC frame section and steel mold release. (**C**) Placement of prepared NGCs in preservation medium. (**D**) NGCs in the stock medium during soy protein release.

**Figure 2 polymers-16-00216-f002:**
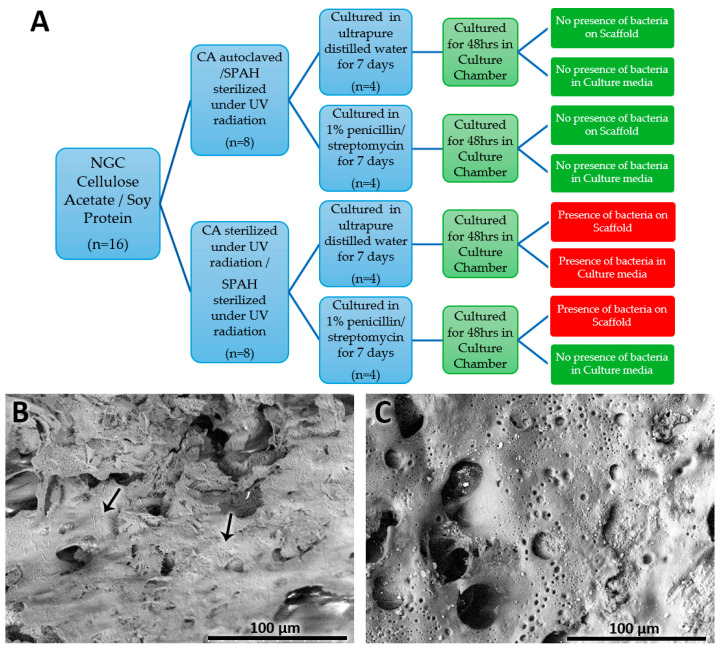
(**A**) Scheme of bacterial growth results from different manufacturing and storage protocols for cellulose and soy protein NGCs. (**B**) Cellulose and soy protein NGCs prepared without autoclaving cellulose showing the presence of bacteria (arrows) (Mag: ×500). (**C**) NGC prepared by autoclaving cellulose acetate and using UV light revealing the absence of bacteria (Mag: ×500).

**Figure 3 polymers-16-00216-f003:**
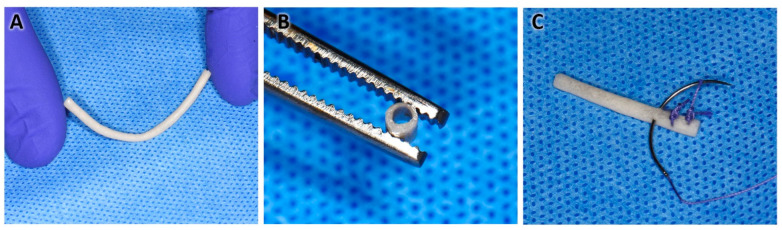
Macroscopic structure of the NGC. (**A**) Flexibility of the NGC while maintaining the tubular characteristic. (**B**) Cross-section of the NGC. (**C**) NGC structure sutured with silk and polyglactin.

**Figure 4 polymers-16-00216-f004:**
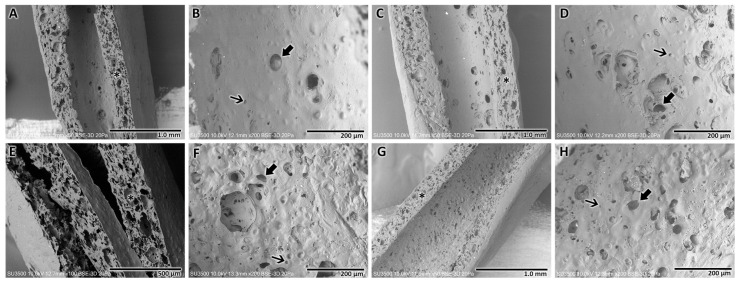
Scanning electron microscopy analysis of NGCs after different forms of sterilization and storage. UV-irradiated CA and SPAH NGC stored in ultrapure water. (**A**) Cross-section of the tubular structure revealing a noncompact wall (*) of the tubular structure (Mag.: ×50). (**B**) Internal surface revealing few pores of different sizes, major (arrow), minor (thin arrow), and a smoother surface (Mag.: ×200). UV-irradiated CA and SPAH NGC stored in 1% penicillin/streptomycin. (**C**) Cross-section of the NGC revealing the noncompact wall (*) of the structure (Mag.: ×50). (**D**) Inner surface of NGC, revealing major (arrow) and minor (thin arrow) pores (Mag.: ×200). Autoclaved CA and UV-irradiated SPAH NGC stored in ultrapure water. (**E**) Note the noncompact wall of the tubular NGC (*) (Mag.: ×100). (**F**) Internal surface of the NGC in which a rougher wall and a greater number of pores can be seen (major—arrow and minor pores—thin arrow) (Mag.: ×200). Autoclaved CA and UV-irradiated SPAH NGC stored in 1% penicillin/streptomycin. (**G**) Cross-section of the NGC revealing the wall with spaces and trabecular structure (*) (Mag.: ×50). (**H**) NGC internal surface with the presence of pores of different sizes (major—arrow; minor—thin arrow) distributed over the entire area of the structure (Mag.: ×200).

**Figure 5 polymers-16-00216-f005:**
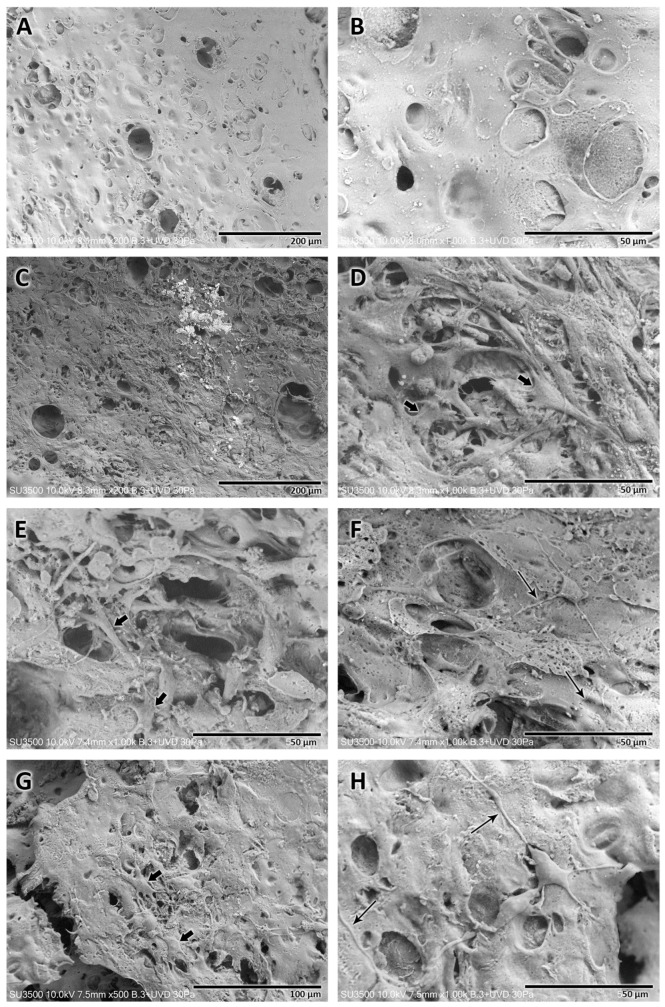
Direct cytocompatibility analysis of Schwann cells seeded on cellulose NGC functionalized with soy protein. (**A**) (Mag.: ×200) and (**B**) (Mag.: ×1000). Internal surface of the NGC without Schwann cell seeding; note the surface with the presence of pores and the absence of bacteria. (**C**) (Mag.: ×200) and (**D**) (Mag.: ×1000). On the inner surface of the NGC after 1 day of Schwann cell seeding, a high cell presence is observed forming a monolayer adhered to a star-shaped structure (arrow). (**E**) (Mag.: ×1000) and (**F**) (Mag.: ×1000). Internal surface of the NGC after 3 days of Schwann cell seeding. Note the presence of adhered Schwann cells (arrow) with a star-shaped shape in smaller quantities compared to the previous period and long cytoplasmic prolongations (long arrow). (**G**) (Mag.: ×500) and (**H**) (Mag.: ×1000). Internal surface of the NGC after 5 days of Schwann cell seeding. The presence of cells with characteristics similar to those observed after 3 days is observed, with the presence of cells with stellate morphology (arrow) and long cytoplasmic extensions (long arrow).

**Figure 6 polymers-16-00216-f006:**
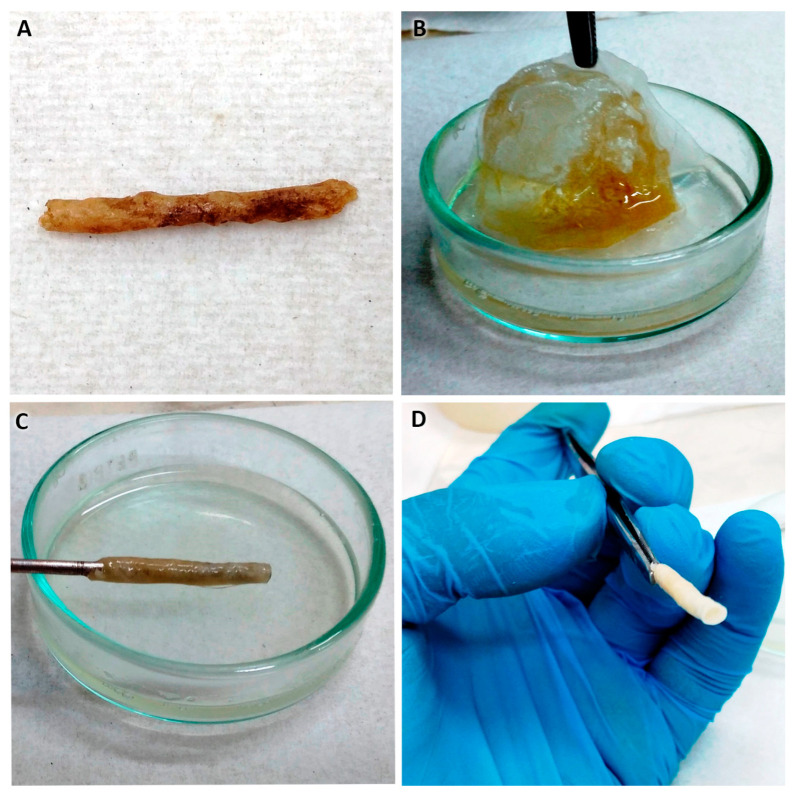
(**A**) NGC manufactured using cellulose acetate and soy protein by solvent casting method. (**B**–**D**) NGC manufactured using cellulose acetate and soy protein using the rolling layer method.

**Table 1 polymers-16-00216-t001:** NGC porosity and surface area.

S1. CA 4.1 and SPAH 16.3 wt%	S2. CA 5.1 and SPAH 13.6 wt%	S3. CA 6.5 and SPAH 13.6 wt%
Pore Volume = 0.041 cc/g	Pore Volume = 0.132 cc/g	Pore Volume = 0.074 cc/g
Pore Diameter Dv(d) = 3.251 nm	Pore Diameter Dv(d) = 3.249 nm	Pore Diameter Dv(d) = 3.28 nm
Surface Area = 40.63 m^2^/g	Surface Area = 146.4 m^2^/g	Surface Area = 76.2 m^2^/g

**Table 2 polymers-16-00216-t002:** NGC wall width after different sterilization processes and storage methods.

Material (Sterilization)	Storage Method	Wall Width (µm, Mean ± SD)	Range (µm)	
			Min.	Max.
CA (UV rad.); SPAH (UV rad.)	ultrapure water	256.4 ± 17.1	237.5	278.1
CA (UV rad.); SPAH (UV rad.)	1% penicillin/streptomycin	267 ± 6.6	260.4	275.8
CA (autoclave); SPAH (UV rad.)	ultrapure water	248.5 ± 1.8	246.7	250.9
CA (autoclave); SPAH (UV rad.)	1% penicillin/streptomycin	259.6 ± 12.1	249	273.6

**Table 3 polymers-16-00216-t003:** NGC internal major pores (>25 μm) after different sterilization processes and storage methods.

Material (Sterilization)	Storage Method	Major Pore Size (µm, Mean ± SD)	Range (µm)	
			Min.	Max.
CA (UV rad.); SPAH (UV rad.)	ultrapure water	61.1 ± 5.4 a	54	66.6
CA (UV rad.); SPAH (UV rad.)	1% penicillin/streptomycin	32.8 ± 3.7 b	28.3	37.7
CA (autoclave); SPAH (UV rad.)	ultrapure water	42.1 ± 4.8	33.5	48.5
CA (autoclave); SPAH (UV rad.)	1% penicillin/streptomycin	40.7 ± 10.1	27.4	60.1

a = CA (UV rad.); SPAH (UV rad.) in ultrapure water larger than other 3 samples. b = CA (UV rad.); SPAH (UV rad.) in 1% penicillin/streptomycin smaller than CA (autoclave); SPAH (UV rad.) in ultrapure water.

**Table 4 polymers-16-00216-t004:** NGC internal minor pores (<25 μm) after different sterilization processes and storage methods.

Material (Sterilization)	Storage Method	Minor Pore Size (µm, Mean ± SD)	Range (µm)	
			Min.	Max.
CA (UV rad.); SPAH (UV rad.)	ultrapure water	11 ± 4.4	5.2	22.1
CA (UV rad.); SPAH (UV rad.)	1% penicillin/streptomycin	9.2 ± 5.5	1.9	23.5
CA (autoclave); SPAH (UV rad.)	ultrapure water	8.5 ± 4.3	2	23.3
CA (autoclave); SPAH (UV rad.)	1% penicillin/streptomycin	9.2 ± 4.8	2.5	23.3

**Table 5 polymers-16-00216-t005:** MTT assay for Schwann cell biocompatibility replicate 1.

Replicate 1 24–25 May 2023	Observance 24 h (Arbitrary Units)	Observance 48 h (Arbitrary Units)	Viability (obs 24 h/obs 48 h)
Sample 1	0.0291	0.0216	74.23%
Sample 2	0.0062	0.0055	88.71%
Sample 3	0.0415	0.0257	61.93%
Mean	0.026	0.018	68.75%

**Table 6 polymers-16-00216-t006:** MTT assay for Schwann cell biocompatibility replicate 2.

Replicate 2 31 May–1 June 2023	Observance 24 h (Arbitrary Units)	Observance 48 h (Arbitrary Units)	Viability (obs 24 h/obs 48 h)
Sample 1	0.0479	0.0367	76.62%
Sample 2	0.0374	0.0304	81.28%
Sample 3	0.0912	0.0556	60.96%
Mean	0.059	0.041	69.52%

## Data Availability

The data that support the findings of this report are available from the corresponding author [F.J.D.] upon reasonable request.

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
