# Peer review of "Comprehensive Development of a Cellulose Acetate and Soy Protein-Based Scaffold for Nerve Regeneration"

_polymers, 2024, doi:10.3390/polym16020216_

Round 1

Reviewer 1 Report

Comments and Suggestions for Authors

The authors of the article presented a detailed description of the developed process of preparation of biocompatible tubular scaffolds for nerve regeneration. All stages of the process are described, different variants of their implementation are analyzed and the optimality of the technology chosen by the authors is substantiated.

The presented material will undoubtedly be interesting and useful for all readers working in the important field of development and manufacturing of new materials for biotechnology.

The article deserves to be published in the journal.

Only small recommendations for the authors to finalize the manuscript can be presented.

1. It is recommended that the authors re-edit the text of the article to identify and correct minor errors and inconsistencies found in the submitted version. One of the examples:

The same electron microscope SEM-Hitachi SU3500 was mentioned as “transmission electron microscope” (line 170) and as “scanning electron microscope” (line 177).

2. The authors use solutions of cellulose-acetate and soy protein in acetone for the manufacture of scaffolds for nerve regeneration. It would be reasonable to provide data indicating the completeness of removal of this injurious solvent from the material of scaffolds during their further processing.

3. The authors claim to have obtained non-brittle, strong scaffolds capable of withstanding mechanical loads. To prove this claim, it would be advisable to conduct mechanical tests of the obtained material and present in the article the results of these tests.

Comments on the Quality of English Language

Minor editing of English language required

Author Response

The authors of the article presented a detailed description of the developed process of preparation of biocompatible tubular scaffolds for nerve regeneration. All stages of the process are described, different variants of their implementation are analyzed and the optimality of the technology chosen by the authors is substantiated.

The presented material will undoubtedly be interesting and useful for all readers working in the important field of development and manufacturing of new materials for biotechnology.

The article deserves to be published in the journal.

Reply: We appreciate the positive comments on our manuscript. We consider all suggestions sent to improve the quality of the publication.

Only small recommendations for the authors to finalize the manuscript can be presented.

  1. It is recommended that the authors re-edit the text of the article to identify and correct minor errors and inconsistencies found in the submitted version. One of the examples:

The same electron microscope SEM-Hitachi SU3500 was mentioned as “transmission electron microscope” (line 170) and as “scanning electron microscope” (line 177).

Reply: Thank you for your observation. In fact, the type of electron microscopy used was poorly indicated. Additionally, we found a few more typos in the cellulose type that were corrected as well.

  1. The authors use solutions of cellulose-acetate and soy protein in acetone for the manufacture of scaffolds for nerve regeneration. It would be reasonable to provide data indicating the completeness of removal of this injurious solvent from the material of scaffolds during their further processing.

Reply: Thank you for your observation. The preparation of the tubular scaffold using the proposed method of solvent evaporation with dip-coating completely eliminates the solvent during preparation. The sentence was rewritten to make clear the elimination of the solvent in the manufacturing process.

  1. The authors claim to have obtained non-brittle, strong scaffolds capable of withstanding mechanical loads. To prove this claim, it would be advisable to conduct mechanical tests of the obtained material and present in the article the results of these tests.

Reply: Thank you for your observation. The authors completely agree that mechanical tests are necessary to prove and mainly quantify the mechanical characteristics of the prepared tubular scaffolds. However, at the moment we do not have the equipment to carry out these tests. The objective of the present study was to describe the difficulties and solutions encountered in the development of a tubular scaffold with the appropriate structural characteristics for its implantation in rat nerve lesions. Additionally with the initial direct biocompatibility tests of Schwann cells. In the future, we will consider the mechanical analyses, as well as the physicochemical analyzes of these developed scaffolds. The lack of mechanical tests was included as a limitation of the study and in the future steps to follow with the development of this nervous scaffold.

Reviewer 2 Report

Comments and Suggestions for Authors

The information presented in the manuscript, Uncomplicated and comprehensive development of a biocompatible Tubular Scaffold for nerve regeneration of cellulose-acetate functionalized with soy protein, is related to the bioengineering of peripheral nerve regeneration. This manuscript contains sufficiently new information, and information is presented in a logical order. Minor proofreading is needed further to improve the readability and quality of the manuscript.

1. Title can be written in short form, such as Cellulose acetate and soy protein-based comprehensive development of a biocompatible tubular scaffold for nerve regeneration.

2. In the abstract section, soy protein is abbreviated as SPAH.  This term can be described according to the nature of the protein, e.g., soy protein acid hydrolysate (SPAH).

3. Schwann cells (SCL 31 4.1/F7) biocompatibility was evaluated by MTT-assay and direct seeding. There may be grammar mistakes in this sentence because, herein, the author checked the biocompatibility of nerve guide conduct. This sentence can be written as biocompatibility of cell-seeded nerve guide conduct was evaluated by MTT-assay.

4. Many words in this manuscript are written in Capital in the middle of sentence structure, and there are also mistakes in unit expression; please carefully check the typing errors. which We can identify 3 different methods (page 2, line#76); biocompatible Tubular Scaffold for nerve; coating method, Molding method (solvent casting) and Rolling method; association Cellulose/isolated soy protein; Scanning Electron Microscopy; voltage of 10Kv (Kv should be written as kV); 5% CO2 (can be written as CO2); please use uniform unit either ml or mL; proposed in the present studio generated a;

5. Sub-title, Biocompatibility of Schwann cells can be written as Biocompatibility evaluation of tubular scaffold.

6. The analysis of cytocompatibility of Schwann cells seeded on cellulose NGC using SEM is not an accurate assay. Most studies used confocal-based assays. If possible, please use confocal-based assays for more clear observation.

Author Response

The information presented in the manuscript, Uncomplicated and comprehensive development of a biocompatible Tubular Scaffold for nerve regeneration of cellulose-acetate functionalized with soy protein, is related to the bioengineering of peripheral nerve regeneration. This manuscript contains sufficiently new information, and information is presented in a logical order. Minor proofreading is needed further to improve the readability and quality of the manuscript.

Reply: We appreciate your general comments on our manuscript. All suggested modifications were analyzed and carried out within the possibility.

  1. Title can be written in short form, such as Cellulose acetate and soy protein-based comprehensive development of a biocompatible tubular scaffold for nerve regeneration.

Reply: Thank you for your observation. The title of the manuscript was simplified as indicated.

  1. In the abstract section, soy protein is abbreviated as SPAH.  This term can be described according to the nature of the protein, e.g., soy protein acid hydrolysate (SPAH).

Reply: We appreciate the comment. The information has been corrected in the summary as indicated.

  1. Schwann cells (SCL 31 4.1/F7) biocompatibility was evaluated by MTT-assay and direct seeding. There may be grammar mistakes in this sentence because, herein, the author checked the biocompatibility of nerve guide conduct. This sentence can be written as biocompatibility of cell-seeded nerve guide conduct was evaluated by MTT-assay.

Reply: We appreciate the comment. The abstract methodology sentence was corrected as indicated.

  1. Many words in this manuscript are written in Capital in the middle of sentence structure, and there are also mistakes in unit expression; please carefully check the typing errors. which We can identify 3 different methods (page 2, line#76); biocompatible Tubular Scaffold for nerve; coating method, Molding method (solvent casting) and Rolling method; association Cellulose/isolated soy protein; Scanning Electron Microscopy; voltage of 10Kv (Kv should be written as kV); 5% CO2 (can be written as CO2); please use uniform unit either ml or mL; proposed in the present studio generated a;

Reply: Thank you for your observation. In fact, throughout the manuscript words were written with capital letters in the middle of the sentence. The entire manuscript was reviewed and corrections were made.

  1. Sub-title, Biocompatibility of Schwann cells can be written as Biocompatibility evaluation of tubular scaffold.

Reply: Thank you for your observation. The subtitle of the discussion has been corrected as indicated.

  1. The analysis of cytocompatibility of Schwann cells seeded on cellulose NGC using SEM is not an accurate assay. Most studies used confocal-based assays. If possible, please use confocal-based assays for more clear observation.

Reply: We appreciate the comment. We agree that confocal microscope analysis would provide better information about the direct biocompatibility of NGC with Schwann cells. However, we currently do not have the necessary inputs to carry out these types of analysis. In the future, our research group is considering performing cell interaction analyzes with confocal microscopy. The lack of this analysis was included as a limitation of our study.

Reviewer 3 Report

Comments and Suggestions for Authors

The authors proposed a simple method of producing NGCs  of cellulose acetate (CA) functionalized with the soy protein (SPAH) for nerve regeneration. However, the experimental results are not enough to support the conclusions. There are several issues should be addressed.

1. The mechanical properties of NGCs should be evaluated.

2. The cell proliferation should be evaluated in detail. 

3. The scale bar should be given properly. 

Author Response

The authors proposed a simple method of producing NGCs of cellulose acetate (CA) functionalized with the soy protein (SPAH) for nerve regeneration. However, the experimental results are not enough to support the conclusions. There are several issues should be addressed.

Reply: We appreciate the comment. Conclusions in the abstract and body of the manuscript were modified based on the results.

  1. The mechanical properties of NGCs should be evaluated.

Reply: The authors thank you for your comment and fully agree with your comment. Mechanical analyzes will allow us to better understand and mainly quantify the properties of the manufactured scaffold. However, we do not have the necessary equipment to carry out these analyses. In the future we will look for ways to make these analyzes viable. The aim of the present study was to explain all the steps and challenges to be able to prepare a tubular scaffold with the appropriate dimensions and structure for implantation in nerve injuries in rats. In addition to delivering an initial structural and biocompatibility analysis. We recognize that there is still a lot of testing to better understand and improve the developed scaffold. The lack of mechanical analyzes was included as a limitation of the study in the discussion section.

  1. The cell proliferation should be evaluated in detail. 

Reply: We appreciate the comment. Once again, we recognize that the manufactured scaffold requires further analysis of both mechanics and cellular biocompatibility. We currently do not have all the inputs and materials for cell proliferation analysis, but we are already planning the purchase of these inputs and the future standardization of these analysis protocols. In the future, our research group will seek to implement more detailed cellular analyses. The need to improve cell compatibility analyzes was included as a limitation of the study.

  1. The scale bar should be given properly. 

Reply: We appreciate the comment. The bars of the SEM images have been modified throughout the manuscript to improve their visibility.

Reviewer 4 Report

Comments and Suggestions for Authors

Researchers have developed a new biocompatible nerve guide conduit (NGC) using cellulose acetate (CA) functionalized with soy protein (SPAH). The NGCs were fabricated using a simple dip-coating and evaporation of solvent method. The NGCs were structurally characterized using scanning electron microscopy (SEM). The porosity of the NGCs was analyzed using a degassing method. The biocompatibility of the NGCs was evaluated using Schwann cells (SCL 4.1/F7). The results showed that the CA/SPAH NGCs had a uniform wall thickness, were flexible, suturable, and free of bacteria. The NGCs were also biocompatible with Schwann cells, which adhered to the structure after 5 days. The researchers concluded that the CA/SPAH NGCs have adequate features to be used for peripheral nerve regeneration studies.

Dear authors, thank you for your interesting manuscript. The Introduction chapter is sufficient.

The quality of Figure 8 shall be improved.

The discussion chapter is too extensive and could be compacted without losing the meaning.

I suggest the following improvements to the Discussion chapter:

  1. Provide more specific details about the challenges and solutions encountered during the tube molding process. For example, the authors could discuss the specific reasons why the solvent casting and rolling layer methods were not successful, and how they overcame these challenges with the dip-coating method.
  2. Quantify the effects of different cellulose acetate and soy protein concentrations on the porosity and integrity of the NGCs. This would provide more rigor to the authors' claims about the optimal concentration for NGC manufacturing.
  3. Discuss the potential implications of the syneresis observed during steam sterilization for the long-term stability of the CA/SPAH NGCs. This would help readers to assess the feasibility of using this NGC for clinical applications.
  4. Please discuss the potential use of tubular scaffold for delivering cells or therapeutic agents to bone defects as well as potential use of scaffolds in combination with other biocompatible materials, such as hydrogels or growth factors, to create a more complex and effective microenvironment for cell growth and tissue regeneration. Please reference this paper - Fabrication and In Vitro Characterization of Novel Hydroxyapatite Scaffolds 3D Printed Using Polyvinyl Alcohol as a Thermoplastic Binder - 10.3390/ijms232314870 which is complementary and also provides valuable insights into the development of biocompatible scaffolds for tissue engineering in regenerative dentistry.

There is an error in line 509 “ empty chapter 6. Patents”.

I see various positives of the paper:

This paper describes an advance technique of creating biocomposite tubular scaffold for nervous system regenerative engineering.

To this end, the researchers have employed numerous experimental approaches to assess the scaffold’s geometry, compatibility, and mechanics.

Summa summarum, the results were favorable with the scaffold having prospects as a nerve regenerating agent.

I am recommending for publication after major revision.

It is necessary for the authors to elucidate on what made the process of developing the scaffold challenging and how they handled the unique challenges encountered along the way.

In this regard, the authors should outline the scope and limitations of the study and possible future studies.

To sum it up, the paper is good and educative. The authors proposed a novel strategy for fabrication of nerve-guiding biodegradable and biocompatible tubular scaffold. The paper needs thorough review and modification, but from me is recommended for publication afterwards.

Comments on the Quality of English Language

is fine

Author Response

Researchers have developed a new biocompatible nerve guide conduit (NGC) using cellulose acetate (CA) functionalized with soy protein (SPAH). The NGCs were fabricated using a simple dip-coating and evaporation of solvent method. The NGCs were structurally characterized using scanning electron microscopy (SEM). The porosity of the NGCs was analyzed using a degassing method. The biocompatibility of the NGCs was evaluated using Schwann cells (SCL 4.1/F7). The results showed that the CA/SPAH NGCs had a uniform wall thickness, were flexible, suturable, and free of bacteria. The NGCs were also biocompatible with Schwann cells, which adhered to the structure after 5 days. The researchers concluded that the CA/SPAH NGCs have adequate features to be used for peripheral nerve regeneration studies.

 Dear authors, thank you for your interesting manuscript. The Introduction chapter is sufficient.

The quality of Figure 8 shall be improved.

Reply: We appreciate the general comment on the manuscript. The quality of figure 8 has been improved as indicated.

The discussion chapter is too extensive and could be compacted without losing the meaning.

I suggest the following improvements to the Discussion chapter:

1. Provide more specific details about the challenges and solutions encountered during the tube molding process. For example, the authors could discuss the specific reasons why the solvent casting and rolling layer methods were not successful, and how they overcame these challenges with the dip-coating method.

Reply: We appreciate the comment. More explanations about the challenges in the tubular scaffold molding process were included.

2. Quantify the effects of different cellulose acetate and soy protein concentrations on the porosity and integrity of the NGCs. This would provide more rigor to the authors' claims about the optimal concentration for NGC manufacturing.

Reply: We appreciate the comment. We are analyzing different concentrations of CA and SPAH in the manufacture of NGC. This way we included 2 more concentrations of these materials analyzed by the porosimeter, one with more CA and the other with less CA and more SPAH. These data were briefly discussed. In the future, our group intends to carry out a more complete analysis of the physicochemical characteristics of the manufactured material in another manuscript.

3. Discuss the potential implications of the syneresis observed during steam sterilization for the long-term stability of the CA/SPAH NGCs. This would help readers to assess the feasibility of using this NGC for clinical applications.

Reply: We appreciate the comment. The implications of syneresis and imbibition of materials were briefly discussed in the section on bacteriological control in the material sterilization process.

4. Please discuss the potential use of tubular scaffold for delivering cells or therapeutic agents to bone defects as well as potential use of scaffolds in combination with other biocompatible materials, such as hydrogels or growth factors, to create a more complex and effective microenvironment for cell growth and tissue regeneration. Please reference this paper - Fabrication and In Vitro Characterization of Novel Hydroxyapatite Scaffolds 3D Printed Using Polyvinyl Alcohol as a Thermoplastic Binder - 10.3390/ijms232314870 which is complementary and also provides valuable insights into the development of biocompatible scaffolds for tissue engineering in regenerative dentistry.

Reply: We appreciate the comment. The potential uses of the developed scaffold were briefly stated at the end of the discussion. We would also like to thank you for mentioning the article, it was mentioned in the manuscript as a contribution to cellular biocompatibility analyses.

There is an error in line 509 “empty chapter 6. Patents”.

Reply: Thank you for your observation. Chapter 6. Patents has been deleted.

I see various positives of the paper:

This paper describes an advance technique of creating biocomposite tubular scaffold for nervous system regenerative engineering.

To this end, the researchers have employed numerous experimental approaches to assess the scaffold’s geometry, compatibility, and mechanics.

Summa summarum, the results were favorable with the scaffold having prospects as a nerve regenerating agent.

I am recommending for publication after major revision.

It is necessary for the authors to elucidate on what made the process of developing the scaffold challenging and how they handled the unique challenges encountered along the way.

In this regard, the authors should outline the scope and limitations of the study and possible future studies.

To sum it up, the paper is good and educative. The authors proposed a novel strategy for fabrication of nerve-guiding biodegradable and biocompatible tubular scaffold. The paper needs thorough review and modification, but from me is recommended for publication afterwards.

Reply: The authors would like to thank you once again for your careful general review of our manuscript, highlighting both the weaknesses and strengths of our study. It should be noted that this is a reviewer who really knows the topic of developing biomaterials for application in regenerative biology. The modifications made substantially improved the prepared manuscript.

Round 2

Reviewer 3 Report

Comments and Suggestions for Authors

The revision is not satisfying. The mechanical strength should be tested, also the cell proliferation is necessary, also the definition of NGC is given twice, which is not necessary. 

Comments on the Quality of English Language

The definition of NGC is given twice, which is not necessary. 

Author Response

Reply: The authors appreciate your comment. Once more, we agree that these analyses would enrich the present study.

However, we do not have the necessary resources and equipment to carry out the aforementioned mechanical and cell proliferation analyses. Furthermore, the deadline given for this major review, 10 days, makes any attempt to seek support and/or partnerships to carry out these analyses unfeasible.

Furthermore, we emphasize again that the objective of the present study did not consider the mechanical analysis of the manufactured NGC, and the cellular analyses that we are able to carry out at this moment have already been carried out.

Regarding the duplicity of the NGC definition, the second definition was deleted as indicated.

I hope you can understand our situation and we appreciate your observations that will be considered in our upcoming studies related to the development and testing of NGC and other materials for peripheral nerve regeneration.

Reviewer 4 Report

Comments and Suggestions for Authors

is ok

Comments on the Quality of English Language

moderate changes

Author Response

Reply: We thank you once again for the constructive observations/comments made previously. In addition, a grammatical review of the entire text was carried out with the aim of improving the English level of the manuscript. We hope that this time the article can be accepted for publication.

Round 3

Reviewer 3 Report

Comments and Suggestions for Authors

The revision is satisfying.